# Treatment of Carbapenem-Resistant *Acinetobacter baumannii* in Real Life (T-ACI): A Prospective Single-Center Observational Study

**DOI:** 10.3390/antibiotics13111007

**Published:** 2024-10-25

**Authors:** Antonio Riccardo Buonomo, Riccardo Scotto, Nunzia Esposito, Giulio Viceconte, Nicola Schiano Moriello, Giulia Zumbo, Ilaria Vecchietti, Amedeo Lanzardo, Carmine Iacovazzo, Francesco Curcio, Emanuela Roscetto, Ivan Gentile

**Affiliations:** 1Department of Clinical Medicine and Surgery, Section of Infectious Diseases, University of Naples Federico II, 80131 Naples, Italy; antonioriccardobuonomo@gmail.com (A.R.B.); giulio.viceconte@gmail.com (G.V.); nicola.schianomoriello@unina.it (N.S.M.); giuliazumbo@hotmail.com (G.Z.); vecchietti.ilaria@gmail.com (I.V.); a.lanzardo@gmail.com (A.L.); ivan.gentile@unina.it (I.G.); 2University Hospital “SS Antonio, Biagio e Cesare Arrigo”, 15121 Alessandria, Italy; 3Department of Neurosciences, Reproductive and Odontostomatological Sciences, University of Naples Federico II, 80131 Naples, Italy; carmine.iacovazzo@unina.it; 4Department of Translational Medical Sciences, University of Naples Federico II, 80138 Naples, Italy; francesco.curcio@unina.it; 5Department Molecular Medicine and Medical Biotechnology, University of Naples Federico II, 80131 Naples, Italy; emanuela.roscetto@unina.it

**Keywords:** *Acinetobacter baumannii*, CRAB, cefiderocol, colistin

## Abstract

**Background:** Carbapenem-resistant *Acinetobacter baumannii* (CRAB) poses significant challenges in healthcare due to its multidrug resistance and high mortality rates among critically ill patients. **Results:** We enrolled 45 patients. Cefiderocol was administered to 40% of patients, often (38.8%) in combination with other antibiotics. Colistin was administered to 60% of patients and always in combination, mostly with ampicillin–sulbactam. The overall ECS and OCS rates were 77.8% and 66.7%, respectively. Patients treated with an initial cefiderocol-based regimen showed a higher rate of ECS compared with patients initially treated with colistin-based regimens (100% vs. 63%, *p* < 0.05). Patients treated with cefiderocol alone showed a higher rate of ECS compared with patients treated with cefiderocol-based regimens (100% vs. 70.6%, *p* < 0.05). No differences in OCS rates were recorded depending on the treatment received. Additionally, cefiderocol regimens were associated with fewer ADRs compared to colistin-based treatment. **Methods:** This prospective observational study enrolled patients with CRAB infections from January 2022 to August 2023. Patients were treated with cefiderocol-based or colistin-based regimens and were monitored for 28 days to assess early clinical success (ECS), overall clinical success (OCS) and adverse drug reactions (ADRs). **Conclusions:** This study highlights the potential advantages of cefiderocol, even used as a monotherapy, in treating CRAB, especially when early clinical and laboratory response was assessed. This research contributes to the ongoing discussion on the most effective and safe treatments for combating CRAB infections, supporting the use of cefiderocol in clinical practice.

## 1. Introduction

*Acinetobacter baumannii* is a pleomorphic, Gram-negative coccobacillus that is strictly aerobic, catalase-positive, oxidase-negative, non-fermenting and non-motile. Its distinctive feature is its ability to tolerate stressful environments and persist on surfaces for extended periods of time. This resilience enables it to survive and spread as a nosocomial pathogen, particularly among critically ill patients, thereby contributing to increased morbidity and mortality [1]. Despite the innate mechanisms of antibiotic resistance conferred by genes in Class-C such as A. *baumannii*-cephalosporinases (ABC), Amp-C, and in Class-D such as OXA-enzyme, *A. baumannii* can develop antibiotic resistance through various mechanisms. These include horizontal gene transfer (HGT) involving the transfer of plasmids, acquisition of mutations, or transpositions of genetic elements (such as the insertion sequence *A. baumannii* ISAba-type element) that modulate the expression of constitutive genes encoding enzymes (such as the efflux system and outer membrane porins) [2]. Furthermore, the unique characteristics of this microorganism are associated with its virulence factors. These include outer membrane proteins (such as porins), cell envelope factors, quorum sensing, biofilm formation, twitching motility via type IV pili, and micronutrient acquisition systems (such as siderophores and iron transporters FecA and FecI) and type II and type VI protein secretion systems [3,4].

*A. baumannii* accounts for up to 10% of all Gram-negative bacterial infections in intensive care units across Europe and the USA. Healthcare-associated infections by this bacterium have been linked with mortality rates ranging from 5% in general hospital wards to 54% in ICUs [5,6]. *A. baumannii* has quickly gained notoriety due to its capacity to develop resistance to multiple antibiotics, thereby rendering many conventional treatment options ineffective. The World Health Organization (WHO) has classified *A. baumannii* as a critical-priority pathogen that poses a significant threat to human health, and for which new antibiotics are urgently needed [7]. Carbapenem resistance was selected as a marker because it is typically associated with a broad range of co-resistance to other antibiotic classes [3].

According to the IDSA guidance on the treatment of antimicrobial-resistant Gram-negative infections, there is no clear “standard of care” antibiotic regimen for *A. baumannii* carbapenem-resistant (CRAB) infections. The general approach for treating infections caused by CRAB involves combination therapy with high-dose ampicillin–sulbactam (with a total daily dose of 6–9 g of the sulbactam component), even when resistance to this agent has been demonstrated, in combination with at least one other agent: polymyxin B, minocycline, tigecycline or cefiderocol [8]. Alternatively, ESCMID guidelines for the treatment of infections caused by multidrug-resistant Gram-negative bacilli suggest a combination therapy including two in vitro active antibiotics among the available antibiotics (polymyxin, aminoglycoside, tigecycline, sulbactam combination) for patients with severe and high-risk CRAB infections, and conditionally recommend against cefiderocol [9]. This recommendation is based on the CREDIBLE RCT, in which the 28-day mortality rates were higher in the cefiderocol group compared with the best-available-therapy group [10]. Since the publication of the CREDIBLE RCT, numerous observational studies, mostly retrospective, have been conducted to evaluate the use of cefiderocol in CRAB infections. The results of these studies have led to a re-evaluation of this antibiotic, demonstrating even lower mortality in the cefiderocol group [11].

Based on this knowledge, we conducted a prospective single-center study (T-ACI: Treatment of *Acinetobacter baumannii* Infections) comparing cefiderocol-based and colistin-based combination therapies for the treatment of CRAB in terms of mortality, hospital discharge and microbiological cure.

## 2. Results

A total of forty-five patients with documented infections caused by CRAB were enrolled in the study. The majority of these patients were male, and a significant proportion (68.9%) were aged 65 years or older. Many of these patients had risk factors for infections caused by multidrug-resistant (MDR) pathogens (Table 1).

At the time of enrollment, most patients (35 out of 45, or 77.8%) were suffering from a life-threatening infection. Of these, 17 (37.8%) presented with sepsis and 2 (4.4%) with septic shock. A significant proportion of patients (25 out of 45, or 55.6%) tested positive for carbapenem-resistant *Acinetobacter baumannii* (CRAB) via a rectal swab. The majority of the isolated CRAB were susceptible to colistin (44 out of 45, or 97.8%), while only a few showed susceptibility to cotrimoxazole (1 out of 45, or 2.2%) or aminoglycosides (2 out of 45, or 4.4%).

All patients were treated with cefiderocol or with colistin, with or without other antibiotics. Cefiderocol was administered to 18 patients (40%), either as a monotherapy (61.1%) or in combination with other antibiotics (ampicillin–sulbactam: 5 patients, 27.8%; fosfomycin: 1 patient, 5.5%; tigecycline: 1 patient, 5.5%).

Colistin was administered to 27 patients (60%) and always in association with other antibiotics (ampicillin–sulbactam: 22 patients, 48.9%; fosfomycin: 3 patients, 6.7%; tigecycline: 2 patients, 4.4%) (Figure 1).

Among the patients with pneumonia (either hospital-acquired pneumonia or ventilator-associated pneumonia), 10/21 (47.6%) were treated with a cefiderocol-based regimen and 11/21 (53.4%) with a colistin-based regimen. Among the patients treated with a cefiderocol-based regimen, seven (70%) received cefiderocol alone and three (30%) received cefiderocol plus ampicillin–sulbactam. All patients treated with a colistin-based regimen received colistin plus ampicillin–sulbactam.

Out of 45 patients, 10 (22.2%) switched their antibiotic regimen due to a lack of clinical–laboratory response within 72 h. The treatment combinations administered after the switch were as follows: cefiderocol monotherapy (5 out of 10, or 50%), cefiderocol combined with ampicillin–sulbactam (3 out of 10, or 30%) and cefiderocol combined with fosfomycin (2 out of 10, or 20%) (Appendix A).

When considering both the initial and switched treatment regimens, 28 patients (62.2%) received cefiderocol and 27 (60.0%) received colistin. Ampicillin–sulbactam was associated with the backbone in 30 (66.7%) patients. The median duration of treatment, including the duration after the antibiotic switch, was 16 days (IQR: 11–25). There were no recorded differences in treatment duration between patients treated with a cefiderocol-based regimen at baseline and those treated with a colistin-based regimen (15 days, IQR: 11 + 23 vs. 16 days, IQR: 11–25M; *p* = 0.73).

Among the 21 patients with hospital-acquired pneumonia (HAP) or ventilator-associated pneumonia (VAP), only 2 (9.5%) switched their antibiotic regimen. Both were initially treated with a combination of colistin and ampicillin–sulbactam, and both switched to cefiderocol monotherapy. Patients with HAP or VAP tended to have a shorter antibiotic treatment duration compared to patients without pneumonia (14 days, IQR: 11–21 vs. 17 days, IQR: 14–27, *p* = 0.09).

The overall ECS and OCS rates were 77.8% and 66.7%, respectively. The overall mortality was 42.2%, while mortality after 28 days from treatment initiation was 35.6%. The median time of exitus occurrence after treatment initiation was 14 days (IQR: 11–25). All patients initially treated with a cefiderocol-based regimen achieved ECS, while it was achieved by 63.7% of patients initially treated with a colistin-based regimen (*p* < 0.05). There was no difference in the ECS rate in patients treated with or without the association of ampicillin–sulbactam with the backbone (Table 2).

At day 28, 24.4% of patients were discharged, while 33.3% died, with an OCS rate of 66.7%. There were no recorded differences in the OCS rate depending on the treatment received. Computing first-line antibiotic regimens and the regimens chosen after antibiotic switches together, patients treated with a cefiderocol-based regimen showed a similar OCR rate compared with patients treated with a colistin-based regimen. There were no differences in the outcomes at day 28 between patients treated with cefiderocol monotherapy and those treated with cefiderocol-based combination regimens.

When comparing the outcomes between patients with pneumonia (both HAP and VAP) and those with other infections, there was a tendency toward higher ECS in patients with pneumonia (54.3% vs. 45.7%, *p* = 0.55). However, patients with pneumonia showed a lower OCS rate compared with those with other infections (33.3% vs. 66.7%, *p* < 0.05). In particular, patients with pneumonia had a significantly higher mortality rate compared to patients with other infections (11 out of 21, or 52.4% vs. 4 out of 24, or 16.7%; *p* < 0.05).

Out of all the enrolled patients, four (8.9%) experienced a recurrence of the infection within 30 days. Half of these patients (two out of four, or 50%) had been treated with a combination of cefiderocol and ampicillin-sulbactam, one (25%) with cefiderocol alone, and one (25%) with a combination of colistin and fosfomycin.

A total of nine adverse drug reactions (ADRs) occurred among eight patients (17.8%). These included respiratory symptoms (four patients, 8.9%), acute kidney injury (two patients, 4.4%), gastrointestinal disorders (one patient, 2.2%), urticaria (one patient, 2.2%) and angioedema without anaphylactic shock (one patient, 2.2%). Most patients with ADRs (five out of eight, or 62.5%) were initially treated with a combination of colistin and ampicillin–sulbactam. Among these, three had respiratory symptoms and two had acute kidney injury. Three patients with ADRs (37.5%) were treated with cefiderocol alone (two patients) or with a combination of cefiderocol and ampicillin–sulbactam. The former group experienced respiratory symptoms (one patient) and gastrointestinal disorders (one patient), while the patient treated with a combination of cefiderocol and ampicillin–sulbactam developed urticaria.

## 3. Discussion

Infections caused by carbapenem-resistant *Acinetobacter baumannii* (CRAB) continue to pose a significant challenge for patients and physicians worldwide due to limited treatment options. Various antibiotic combinations have been proposed and utilized in recent years, with colistin being a common component in most of these combinations. Colistin, however, is a difficult-to-manage drug and can lead to potentially severe adverse drug reactions (ADRs), primarily acute kidney injury and neurotoxicity [12]. However, until several years ago, colistin-free combinations did not achieve satisfactory effectiveness against CRAB [13,14]. The introduction of cefiderocol into clinical practice was therefore met with great enthusiasm by the international medical community. Several studies have indeed demonstrated the in vitro activity of cefiderocol against CRAB [15]. However, the phase 3 randomized controlled trial CREDIBLE-CR reported a higher mortality rate in patients with CRAB infections treated with cefiderocol compared with patients treated with the best available therapy [16]. This difference was attributed to several factors, one of which was the very low life expectancy of patients with CRAB infections included in the study. The contrasting evidence between in vitro studies and the CREDIBLE-CR trial has prompted the development of several spontaneous real-world studies to test the effectiveness of cefiderocol against CRAB infections.

Our study is a real-world observational prospective study conducted at a single university center. It aimed to compare the effectiveness of cefiderocol-based treatment regimens in terms of mortality, hospital discharge and microbiological eradication. We classified the treatments into two main groups: cefiderocol-based and colistin-based. The majority (77.7%) of the 45 patients enrolled had a life-threatening infection. Nevertheless, the overall rates of early clinical success (ECS) and overall clinical success (OCS) were relatively high, being 77.8% and 66.7%, respectively. Patients who received a cefiderocol-based regimen demonstrated a significantly higher rate of ECS compared to those treated with a colistin-based regimen. Remarkably, all patients treated with cefiderocol, whether alone or in combination, achieved ECS. The addition of ampicillin–sulbactam to the primary treatment (cefiderocol or colistin) did not lead to a significant increase in ECS rates. However, despite the difference in ECS rates according to the initial treatment regimen, no differences were observed in OCS rates when considering both the initial treatment regimen and overall treatment regimen.

Our data differ from what emerged from the CREDIBLE-CR trial, in which patients treated with cefiderocol had higher mortality when compared to the best available therapy [16]. However, data from other observational studies align with our results. For instance, Oliva et al. showed, in a cohort of 106 patients, significant differences in clinical cures between cefiderocol-based or non-cefiderocol-based treatments for CRAB infections in COVID-19 patients, 66% vs. 44.4% (*p* = 0.027). However, no difference in mortality emerged in this study (*p* = 0.492) [17]. In another Italian study by Falcone et al., an advantage in terms of mortality in patients treated with cefiderocol (vs colistin) for CRAB bloodstream infections was demonstrated [10]. Onorato et al. actually showed, in a systematic meta-analysis, 42% mortality in patients treated with cefiderocol vs. 60% in patients treated with the best available therapy (*p* = 0.016) [18]. A recent meta-analysis by Zhan Y et al., encompassing six observational studies, revealed that cefiderocol treatment for severe CRAB infections is associated with a lower all-cause mortality rate compared to colistin-based regimens (RR = 0.71, 95% CI: 0.54–0.92, *p* = 0.01) and a reduced 30-day mortality rate (RR = 0.64, 95% CI: 0.43–0.95, *p* = 0.03) [19]. Therefore, accumulating evidence, along with the results of our study, argues against the use of colistin as a first-line therapy for CRAB infections. Moreover, our findings highlight the efficacy of cefiderocol, even when used as a monotherapy. We acknowledge that no definitive conclusion about the efficacy of a monotherapy can be drawn based on our results and that the optimal treatment for CRAB infection may likely vary depending on the site of the infection. For example, the data on cefiderocol concentrations in epithelial lung fluid (ELF) are controversial, with some pharmacokinetic studies suggesting that ELF concentrations may be insufficient against bugs with high MICs to the drug [20]. In our study, patients affected by HAP or VAP showed a tendency have higher rates of OCS compared with patients with other infections. They were treated with a combination of ampicillin–sulbactam with either cefiderocol or colistin (77.7%) or cefiderocol alone (33%). According to a recent multi-center study of 73 bacteremic VAP cases caused by CRAB, Russo et al. showed significant 14-day and 30-day mortality reductions when cefiderocol was combined with fosfomycin compared to patients treated with colistin and fosfomycin [21]. Therefore, we can postulate that the optimal treatment for CRAB infection should also be evaluated by considering the site of infection, and in pneumonia, combination treatments could be considered the treatment of choice. Further studies are needed to confirm this hypothesis.

The main limitation of our study was its limited sample size but, different from many other real-life studies, it had the advantage of a prospective design. Moreover, antibiotic therapy in enrolled patients was prescribed only by four different physicians, thus limiting the spectrum of prescribed drugs and combinations. In other real-life studies, there was indeed a high variability in prescribed combination regimens and indications for monotherapy. Furthermore, all patients had documented infections caused by carbapenem-resistant *Acinetobacter baumannii* strains.

In conclusion, our study found a higher efficacy and safety of cefiderocol, even used as a monotherapy, in terms of early clinical response compared to colistin-based regimens in patients with different types of CRAB infections. These findings should be validated in larger studies to address the current gap in evidence, ideally enrolling patients with a higher life expectancy. This would assist in determining the real need for combination treatment in CRAB infections and in evaluating the optimal therapeutic strategy for each site of infection.

## 4. Methods

We conducted a prospective study enrolling all adult patients diagnosed with Carbapenem-resistant *Acinetobacter baumannii* (CRAB) infections, who were hospitalized at the University of Naples Federico II from 1 January 2022 to 31 August 2023 and received an infectious disease (ID) consultation and treatment. All patients received targeted antibiotic treatment against carbapenem-resistant *A. baumannii*. We excluded patients with rectal or other-site colonization by *A. baumannii* and those who received empirical therapy for CRAB due to worsening clinical conditions and known colonization.

All included patients received monotherapy with cefiderocol (2 g every 8 h, adjusted for renal dosing if necessary) or combination therapy with colistin (9MUI as a loading dose followed by 4.5 MUI twice per day, adjusted for renal dosing if necessary) or cefiderocol plus a second antibiotic active against *Acinetobacter baumannii*: ampicillin–sulbactam (9 g of sulbactam per day), tigecycline (100 mg as a loading dose followed by 50 mg twice per day) or fosfomycin (from 12 g to 24 g per day). The choice of the initial treatment regimen was at the discretion of the individual physician. However, all therapies were prescribed by four ID specialists.

At the time of diagnosis, the presence or absence of life-threatening infections was assessed. Life-threatening infections were defined by the potential for unfavorable evolution, sometimes lethal, from a few hours to a few days. In particular, ventilator-associated pneumonia (VAP), catheter-related bloodstream infection (CR-BSI) and sepsis were considered life-threatening infections. Hospital-acquired pneumonia (HAP) and VAP were defined according to International ERS/ESICM/ESCMID/ALAT guidelines, namely, infections of the pulmonary parenchyma that developed in patients admitted to the hospital for 48 h (HAP) and pneumonia that developed at least 48 h after endotracheal intubation (VAP) [22]. CR-BSI was defined as the positivity of blood cultures obtained through a catheter ≥120 min before those obtained from a peripheral vein with the same microorganism, in patients with fever, chills or new-onset hypotension and/or signs of infection proximal to insertion sites according to IDSA guidelines [23]. Sepsis was defined according to the Surviving Sepsis Campaign 2021 [24] as a difference of 2 points in the Sequential Organ Failure Assessment (SOFA) Score at the onset of infection compared to baseline. Complicated intra-abdominal infection (cIAI) was defined according to the Infectious Diseases Society of America (IDSA) as an abdominal infection that extends beyond the hollow viscus of origin into the peritoneal space and is associated with either abscess formation or peritonitis [25]. Uncomplicated UTI was defined as an infection causing local bladder signs and symptoms without fever, other signs of systemic infection or findings suggestive of kidney involvement. ABSSSI was defined according to the Food and Drug Administration (FDA) definition as cellulitis/erysipelas, major skin abscesses and wound infections, all requiring a minimum lesion surface area of 75 cm^2^ [26]. Endocarditis was defined as cases that included combinations of 2 major criteria, 1 major criterion and at least 3 minor criteria, or 5 minor criteria from the 2023 European Society of Cardiology (ESC) modified diagnostic criteria of infective endocarditis [27]. Osteomyelitis was defined as an infection with radiologically or culturally demonstrated bone involvement. Cardiac implantable electronic device infection was defined as a systemic infection with or without infective endocarditis and with or without local infection according to the 2020 ESC guidelines [28].

Patients were followed up for 28 days, assessing clinical and laboratory parameters and evaluating any possible development of adverse reactions, side effects or therapy failures.

The primary outcome was early clinical success (ECS). ECS was defined by a noticeable clinical response to the initial treatment regimen within 72 h. This response was assessed based on the improvement of infection symptoms (such as fever), along with a decrease in white blood cells, C-reactive protein and procalcitonin. If there was no clinical–laboratory response to therapy within 72 h, a switch to another treatment regimen was made at the clinician’s discretion. The secondary outcome was overall clinical success (OCS). Patients achieved OCS if they were discharged or alive at day 28 after the diagnosis of CRAB infection. For a safety evaluation, adverse drug reactions (ADRs) were monitored for each patient enrolled. A close evaluation of each clinical record was performed to monitor either clinical or laboratory ADR development.

### Statistical Analysis

The Kolmogorov–Smirnov test was utilized to evaluate the distribution type (Gaussian or non-Gaussian) of each variable. Categorical variables were reported as absolute numbers and percentages (*n*, %), while continuous variables were reported as mean ± standard deviation or median ± interquartile range, depending on the type of distribution (parametric or non-parametric, respectively). Comparisons of categorical variables were conducted using the χ^2^ test (or Fischer’s exact test, when appropriate). Comparisons of continuous variables were performed using the t-Student test for parametric distributions or the Mann–Whitney U test for non-parametric distributions. The outcome analysis for ECS was performed considering only first-line treatment. The outcome analysis for OCS was performed by computing both first-line treatments and antibiotic treatments after switching. For all tests, a *p*-value of <0.05 was considered statistically significant. Statistical analysis was performed using IBM SPSS© version 25.0.

## Figures and Tables

**Figure 1 antibiotics-13-01007-f001:**
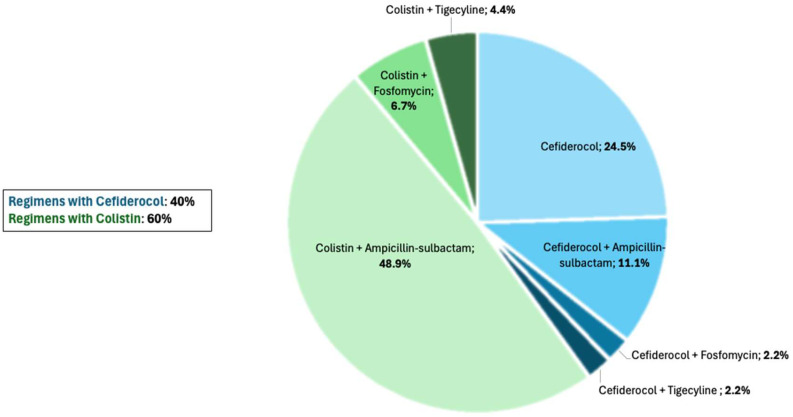
Baseline treatments among enrolled patients.

**Table 1 antibiotics-13-01007-t001:** Baseline characteristics of the enrolled patients (N = 45).

Age (years; median, IQR)	70 (63–77)
Male sex (*n*, %)	33 (73.3)
Ward at admission (*n*, %)	
- Medicine - Surgery - Intensive care unit	21 (46.7) 9 (20.0) 15 (33.3)
Comorbidities (*n*, %)	
- Diabetes mellitus - COPD - Cardiac disease - Solid tumor - Hematological disease - CKD - Liver disease	11 (26.8) 7 (17.1) 9 (22.0) 7 (17.1) 4 (9.8) 7 (17.1) 3 (2.4)
Charlson comorbidity index (median, IQR)	2 (1–5)
Risk factors for infection (within 3 months) (*n*, %)	
- Treatment with antibiotics - Total parenteral nutrition - Previous hospital admissions - Surgery - SARS-CoV-2 infection in the last 30 days	17 (42.5) 4 (8.9) 19 (45.2) 9 (20.0) 21 (46.7)
Site of infection (*n*, %)	
- VAP - HAP - CRBSI - cIAI - UTI - ABSSSI - Endocarditis - Osteomyelitis - Device infection	19 (42.2) 2 (4.4) 14 (31.1) 3 (6.7) 2 (4.4) 7 (15.6) 1 (2.2) 2 (4.4) 1 (2.2)
Life-threatening infection (*n*, %)	35 (77.8)
Sepsis at baseline (*n*, %)	17 (42.9)
Septic shock at baseline (*n*, %)	2 (4.4)

IQR: interquartile range; COPD: chronic obstructive pulmonary disease; CKD: chronic kidney disease; VAP: ventilator-associated pneumonia; HAP: hospital-acquired pneumonia; CRBSI: catheter-related bloodstream infection; cIAI: complicated intra-abdominal infection; UTI: urinary tract infection; ABSSSI: acute bacterial skin and soft structure infection.

**Table 2 antibiotics-13-01007-t002:** Outcome rates at day 28 according to treatment regimen (*n* = 45).

	Early Clinical Success			Overall Clinical Success		
	Yes	No	*p*-Value	Yes	No	*p*-Value
Baseline treatment regimen (*n*, % ^):						
- Cefiderocol-based - Colistin-based	18 (100) 17 (63.0)	0 (0.0) 10 (37.9)	<0.05	13 (72.2) 17 (63.0)	5 (27.8) 22 (37.0)	0.519
- Included ampicillin–sulbactam - Without ampicillin–sulbactam	21 (77.8) 14 (40.0)	6 (22.2) 4 (22.2)	0.647	16 (59.3) 14 (77.8)	11 (40.7) 4 (22.2)	0.197
- Cefiderocol monotherapy - Cefiderocol combination	11 (100) 7(100)	0 (0.0) (0.0)	n.a.	8 (72.7) 5 (71.4)	3 (27.3) 2 (28.6)	0.676
Overall treatment regimen ^$^ (*n*, % ^):						
- Cefiderocol-based - Colistin-based	# #	# #	# #	19 (67.9) 11 (64.7)	9 (32.1) 6 (35.3)	0.828
- Included ampicillin–sulbactam - Without ampicillin–sulbactam	# #	# #	# #	19 (63.3) 11 (73.3)	11 (36.7) 4 (26.7)	0.502
- Cefiderocol monotherapy - Cefiderocol combination	# #	# #	# #	9 (56.3) 10 (83.3)	7 (43.8) 2 (16.7)	0.271

^ Raw percentages. #, ^$^ The overall treatment regimen includes both baseline treatment and treatments after antibiotic switches. For this reason, early clinical success was not assessed. n.a.: not analyzable.

## Data Availability

The research data are stored at the University of Naples Federico II—Department of Clinical Medicine and Surgery. Research datasets are stored by the investigator and available under specific request.

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
