# Peer review of "Treatment of Carbapenem-Resistant Acinetobacter baumannii in Real Life (T-ACI): A Prospective Single-Center Observational Study"

_antibiotics, 2024, doi:10.3390/antibiotics13111007_

Round 1
Reviewer 1 Report
Comments and Suggestions for Authors
Dear authors, I have read your manuscript Treatment of Carbapenem-resistant Acinectobacter baumannii in real-life (T-ACI): a prospective single center observational study, and these are my comments and suggestions:
General: Write "Acinetobacter baumannii" in italics throughout the manuscript.
Title (Lines 1-4): What do you mean by "T-ACI"? I'm not sure it will be clear to the reader. The abbreviation is not used elsewhere in the manuscript.
Lines 28-29: Change to "Patients treated with an initial cefiderocol-based regimen showed a higher rate of ECS..." (it's either "a higher rate" or "higher rates").
Lines 31-32: Change to "No differences in OCS rates ...".
Lines 32-33: This sentence seems to be contradictory to what you said before. Please clarify.
Line 38: Change to "..., supporting the use of cefiderocol in clinical practice."
Lines 282-284: Change to "This difference was attributed to several factors, one of which was the very low life expectancy of the patients with CRAB infections included in the study."
Line 292: Is the abbreviation "ECS" referring to Effective Clinical Success or early clinical success as mentioned in line 24?
Lines 304-306: What kind of differences in clinical cure were these? Please specify.
Line 306: Change "emergent" to "emerged".
Comments on the Quality of English LanguageI have given a few examples.
Author Response
Please find attached a point-by-point response.
Thank you

Reviewer 2 Report
Comments and Suggestions for Authors
Thisi a good work and well written although AI help
First, species names MUST be in italics
Discussion needs to be improved, particularly teh relatioship of the results obatined and other works worldwide, not only form Italy
Figure 1 has to be improved...not feasible to understand
Figure 2 might be submitted as supplementary material
Comments on the Quality of English LanguageMinor revisions are needed
Author Response

(The authors gave the same response as above.)
